# Low Carb and Ketogenic Diets Increase Quality of Life, Physical Performance, Body Composition, and Metabolic Health of Women with Breast Cancer

**DOI:** 10.3390/nu13031029

**Published:** 2021-03-23

**Authors:** Ulrike Kämmerer, Rainer J. Klement, Fabian T. Joos, Marc Sütterlin, Monika Reuss-Borst

**Affiliations:** 1Department of Obstetrics and Gynaecology, University Hospital of Würzburg, D-97080 Würzburg, Germany; 2Department of Radiation Oncology, Leopoldina Hospital Schweinfurt, D-97422 Schweinfurt, Germany; rainer_klement@gmx.de; 3Departement of Anaesthesiology, Regional Clinic Holding RKH GmbH, D-71640 Ludwigsburg, Germany; joos.f@gmx.net; 4University Medical Center Mannheim, Heidelberg University, D-68159 Mannheim, Germany; masusek@gmx.de; 5Center for Rehabilitation and Prevention Bad Bocklet, D-97708 Bad Bocklet, Germany; info@reuss-borst-medizin.de

**Keywords:** breast cancer, rehabilitation, ketogenic diet, low carb diet, supportive care

## Abstract

Breast cancer (BC) patients often ask for a healthy diet. Here, we investigated a healthy standard diet (SD), a low carb diet (LCD), and a ketogenic diet (KD) for BC patients during the rehabilitation phase. KOLIBRI was an open-label non-randomized one-site nutritional intervention trial, combining inpatient and outpatient phases for 20 weeks. Female BC patients (*n* = 152; mean age 51.7 years) could select their diet. Data collected were: Quality of life (QoL), spiroergometry, body composition, and blood parameters. In total 30, 92, and 30 patients started the KD, LCD, and SD, respectively. Of those, 20, 76, and 25 completed the final examination. Patients rated all diets as feasible in daily life. All groups enhanced QoL, body composition, and physical performance. LCD participants showed the most impressive improvement in QoL aspects. KD participants finished with a very good physical performance and muscle/fat ratio. Despite increased cholesterol levels, KD patients had the best triglyceride/high-density lipoprotein (HDL) ratio and homeostatic model assessment of insulin resistance index (HOMA-IR). Most metabolic parameters significantly improved in the LCD group. SD participants ended with remarkably low cholesterol levels but did not improve triglyceride/HDL or HOMA-IR. In conclusion, both well-defined KDs and LCDs are safe and beneficial for BC patients and can be recommended during the rehabilitation phase.

## 1. Introduction

Breast cancer (BC) is one of the most common cancer types worldwide, with over 2 million new cases every year (www.wcrf.org (accessed on 19 March 2021)) [1]. Despite a generally good prognosis, the disease itself as well as its standard treatments via surgery, radiation, and chemotherapy can have a negative influence on the health and physical fitness of affected patients. Typical problems such as weakness and loss of muscle mass (sarcopenia) are on the one hand significantly related to treatment adherence and overall survival [2] and, on the other hand, to the particular cancer patient’s whole-body metabolism. An increased inflammatory environment may characterize the latter [3,4]. Parallel to inflammation, peripheral insulin resistance occurs and leads to a decreased ability of healthy tissue to metabolize glucose for energy demands [5,6]. As compensation, fat oxidation rate increases [7,8].

Hoping to support their healing process and long-term survival, many BC patients ask their health care providers about the possibility of integrating a healthy eating pattern or using web-based diets [9]. In general, physicians and cancer societies advise a healthy standard diet (SD) to normalize body weight, since obesity is known to be negatively associated with breast cancer prognosis [10,11]. Such a SD is low in fat and rich in high-fiber and starchy carbohydrates, fruits, and vegetables [12,13]. Such a SD typically provides at least 50% energy from carbohydrates, 0.8 g/kg body weight protein, and approximately 30% energy from fat [14].

In recent years, “high fat, low carb” diets have been discussed as a metabolically adapted therapy by some clinical nutrition societies [15]. Indeed, preliminary studies have shown that a fat-rich diet is able to protect muscle mass in the presence of catabolic stimuli [16,17,18]. The most stringent nutritional regime, high in fat and low in carbohydrates, is the ketogenic diet (KD). It provides at least 75% of daily calories from fat, is adequate in protein (1.0–1.4 g/kg body weight/day), and is very low in carbohydrates (20–50 g per day). This diet is still a matter of concern and leads to a debate among oncologists and nutritionists, who expect cardiovascular side effects and loss of quality of life due to the high amount of fat [19,20,21].

A less strict but also fat-enriched diet is the LCD (low carbohydrate diet). Its idea is to keep insulin levels low in order to prevent or reduce metabolic syndrome, a pathological metabolic state that has been linked to worsening cancer outcomes [22]. The LCD allows an intake of up to 120 g of carbohydrates per day; it is balanced in protein (20% of energy/day) and rich in fat (remaining calories) [23,24,25].

All three diet types avoid refined sugar, alcohol, and highly processed foods and include higher amounts of fiber and healthy plant-based oils than the average Western diet. Thus, all three diets seem to have the potential to support cancer patients. In this respect, the aim of this open-label trial was to compare the three diet types (SD, KD, and LCD) in BC patients and to assess feasibility, safety, and tolerability. The focus was on quality of life, body composition, physical performance, and serum biochemistry during the rehabilitation phase. Based on what was known from the literature and basic physiology at the time of study conception, it was hypothesized that all diet types are safe but that a lower carbohydrate intake in the KD and LCD groups leads to more favorable changes in body composition and metabolic parameters than in the SD group.

## 2. Materials and Methods

The “Ketogenic or LOGI Diet in a Breast Cancer Rehabilitation Intervention” (KOLIBRI) trial (ClinicalTrials.gov (accessed on 19 March 2021) Identifier: NCT02092753) was approved by the Ethics Committee of the Bavarian Medical Association (Bayerische Landesärztekammer; No. 13082) (“LOGI” stands for “Low Glycemic Index” and is the concept of the LCD used herein). All study participants signed informed consent.

### 2.1. Study Design

This was an open-label non-randomized nutritional intervention trial for 20 weeks consisting of three phases (Figure 1) with three intervention groups in parallel:Three weeks of an inpatient multimodal intervention (for details, see Appendix A) in the rehabilitation center (initial examination T0), followed by the implementation of the allocated diet and training of the patients in diet calculation, cooking, and realization of the diet regimen in routine daily life.Sixteen-week outpatient phase: continuing the selected nutritional regime under close contact and supervision of the study team, accompanied by food diaries and daily urine measurements of ketones (KD group).One closing week of intervention at the rehabilitation center (final examination: T20).

### 2.2. Patients

Over a period of 14 months (July 2015–September 2016), 152 women aged between 26–69 years (mean age 51.7 years) were enrolled in the study during standard rehabilitation at one specific center after the treatment of primary or recurrent BC. All patients gave written informed consent. Baseline characteristics of the patients are shown in Table 1.

Exclusion criteria were Karnofsky Index < 70 and/or expected life span < 12 months, additional malignant tumors at the time of recruitment, participation in other trials, unintentional weight loss and body mass index (BMI) < 18, dementia or other clinically relevant alterations of mental status that could impair the ability to cope with the diet, insufficient knowledge of the German language and therefore inability to follow instructions, type 1 diabetes mellitus, decompensated heart failure (New York Heart Association (NYHA) > 2), myocardial infarction within the last 6 months, symptomatic atrial fibrillation, severe acute infection, pregnancy or pancreatic insufficiency.

### 2.3. Procedure

In Germany, all cancer patients are entitled to attend a three-week inpatient rehabilitation program financed by the German Pension Insurance. Such programs aim to provide patients support in managing the medium- and long-term challenges associated with their condition. The concept of these programs is comprehensive and multiprofessional. Its overarching goal is strengthening patients’ self-management competencies by providing them information, skills, and support.

At the beginning of the rehabilitation program, patients were introduced to the study aims and procedures. Certified dietitians presented the three different diet regimens in a neutral lecture as healthy diet choices. On the morning of day two, all patients who had given written consent to participate underwent primary examination (T0: body weight and size, body composition, serum blood parameters, physical performance, quality of life questionnaires, assessment of individual dietary history) and selected their dietary regime (no randomization), which immediately started with the first lunch. Out of the 152 patients, 30 selected KD, 92 LCD, and 30 SD. At the first evening, one patient switched from KD to LCD and one from LCD to SD. One patient stopped participation at day three, resulting in 29 KD, 92 LCD, and 31 SD patients throughout the three weeks of in-house rehabilitation (Figure 1). This rehabilitation program comprised a structured multimodal treatment program according to the national guidelines for rehabilitation in breast cancer (www.reha-therapiestandards-drv.de (accessed on 19 March 2021)). This program included group psychological counseling and psychoeducation twice a week, relaxation techniques (autogenic training) twice a week, and endurance training (ergometry, Nordic walking) and strength training 2–3 times a week. In the case of lymphedema, lymphatic drainage was performed twice a week. Ergotherapy and physiotherapy were only applied in the case of functional deficits. All patients also received advice on socio-legal issues. In addition to these treatments, extensive nutritional training and advice on the chosen diet according to the study protocol was provided.

The composition and guidelines for the three dietary patterns are given in Appendix A. All dietary compositions were based on a “healthy Mediterranean-like” diet, with emphasis on plant based fat sources low in omega-6 fatty acids.

After 16 weeks of an outpatient phase, patients underwent another week of a “rehabilitation refresher” while maintaining their diet until the final examination. Unfortunately, in five cases, health insurances refused to cover the costs of this week, and therefore four KD patients and one SD patient could not participate in the final examination. Here, telephone interviews revealed that all five persons adhered to their diet until T20. Additional reasons for dropouts included lack of motivation, personal (not medical/health) reasons, and health problems (Figure 1). At T20, there were 20 patients in KD, 76 in LCD, and 25 in SD.

### 2.4. Parameters Analyzed at T0 and T20

At day one of phase 1 (T0) and day 4 of phase 3 (T20), body weight and size were measured after an overnight fast, followed by taking blood and serum samples. Then, bioimpedance analysis (BIA; Nutriguard-M Data-Input GmbH, Pöcking, Germany) and dual-energy X-ray absorptiometry (DXA; Horizon DXA system Explorer S/N 90425, HOLOGIC, Marlborough, MA, USA) were performed according to standard protocols [26,27]. After breakfast, quality of life (QoL) data were assessed via standardized questionnaires (EORTC QLQ-C30 version 3.0), and finally, physical performance was assessed by spiroergometry on a bicycle ergometer and a standard ramp protocol (Ergoline 900 digital, ergoline GmbH, Binz Germany), supervised by the experienced team of the rehabilitation center.

### 2.5. Diet Analysis

Average energy and macronutrient intake per day through the outpatient phase were calculated based on food diaries. Per patient, a trained dietitian analyzed 3–5 randomly selected days with the PRODI 5 program (Nutri-Sciences GmbH, Freiburg, Germany).

Ketosis was documented daily in the KD group by means of urine tests (Ketostix, Bayer, Basel, Switzerland).

### 2.6. Data Collection and Statistical Analysis

A routine clinical lab analyzed blood and serum samples. BIA was analyzed with the Thetis V3.1 software (FORANA GmbH, Frankfurt, Germany), DXA with the QDR for Windows XP 12.5 software (HOLOGIC), and spiroergometry with the SDS104 software (Ganshorn, Niederlauer, Germany). Body composition parameters of interest derived from DXA were bone mineral density (BMD), fat mass (FM), visceral FM, skeletal muscle mass (SMM), and those derived from BIA were body cell mass (BMC), FM, and phase angle at 50 kHz (PA).

Parameter values are given as median (range). Within-group differences between T0 and T20 were evaluated using the Wilcoxon signed-rank test; between-group differences were evaluated using the Kruskal–Wallis test. No adjustments of *p*-values were performed [28], but we used a more stringent threshold of *p* < 0.005 to define statistical significance [29]. Briefly, the classical *p*-value threshold of 0.05 is only consistent with weak evidence against the null hypothesis when converted to minimum Bayes factors [30], which means that such results are often not reproducible. In order to avoid that, and because of the large number of tests performed in this study, a lower threshold of 0.005 was applied, corresponding to strong evidence against the null hypothesis within a likelihood- or Bayes factor-based conception of evidence.

Data were analyzed with R version 3.5.0 and shown as plots using Prism 6.05 (GraphPad Software, San Diego, CA, USA).

## 3. Results

### 3.1. Baseline Characteristics

The baseline characteristics of patients measured at T0 are shown in Table 1. There were significant differences between the groups for body composition and the presence of metastases as well as a non-significant but obvious difference in anti-hormone therapy. Post hoc analyses revealed the KD group to have significantly lower BMI (*p* = 0.0003), FM (*p* = 0.0002), and visceral FM (*p* = 0.0002) and higher metastatic burden (*p* = 0.0006) than the LCD group, while the differences from the SD group missed statistical significance (BMI: *p* = 0.008; FM: *p* = 0.019; visceral FM: *p* = 0.023; metastases: *p* = 0.11). LCD group and SD group were similar in baseline characteristics.

Due to the differences in baseline characteristics and the fact that the majority of KD group participants had prior experience with low carb diets (see Section 3.2), we decided to not perform intergroup difference statistics at T20 but to concentrate on within-group changes instead.

### 3.2. Choice of Diet Type

An analysis of diet history questionnaires revealed that 22 of the 29 patients in the KD group already had experience with consuming a LCD or KD over a time period of 6–24 months. In contrast, patients in the LCD or SD group started from a Western diet. The majority of patients selected LCD. All three diets were easy to implement into the participants’ lives as judged by phone interviews, email exchanges, and questionnaires throughout the outpatient phase as well as personal interviews during their stay in the rehabilitation center. A few patients reported mild headache and digestive problems at the beginning; they were self-resolving and could not clearly be assigned to one diet type.

### 3.3. Energy and Protein Uptake

Since cancer patients need a considerable amount of energy and an increased daily intake of protein [31], both components were analyzed on the basis of the food diaries maintained by the patients in the outpatient phase. Here, KD provided an average of 32.5 ± 1.5 kcal/kg/d, which was significantly higher than the energy supply obtained from the LCD (24.3 ± 0.7 kcal/kg/d; *p* < 0.0001) or SD (26.4 ± 1.2 kcal/kg/d; *p* = 0.0005). Hence, LCD just missed the aspired energy intake of 25–30 kcal/kd/d (Figure 2A). Both KD (1.33 ± 0.07 g/kg/d) and LCD (1.2 ± 0.03 g/kg/d) supplied sufficient average amounts of protein, while the SD group (0.98 ± 0.04 g/kg/d) failed to reach the aspired protein intake of 1.0–1.5 g/kg/d (Figure 2B). Patients in the KD group reached the intended ketogenic ratio (grams of fat divided by grams of carbohydrates plus protein) of 1.6:1 (average 1.65 ± 0.08) and exhibited stable ketosis according to the daily urine measurements (not shown). As expected, LCD patients had a higher ketogenic ratio than SD patients (Figure 2C).

### 3.4. Physical Performance

A total of 19, 70, and 23 patients from the KD, LCD, and SD group completed both baseline and T20 spiroergometry (Table 2). During the intervention, the respiratory quotient (RQ) in the KD group decreased to 0.75 (0.65–0.83), almost reaching the 0.7 value of pure fat oxidation, and was therefore significantly lower than the RQ in the LCD (*p* = 4.4 × 10^−6^) and SD (*p* = 1.3 × 10^−5^) groups. The RQ in both the LCD and SD group remained stable.

All three groups improved their VO_2_/kg (max) values during the intervention, which was significant in both the LCD (*p* = 0.0003) and SD (*p* = 0.0013) group. Despite being the group with the highest percentage of advanced diseases, patients in the KD group performed best in the ergometer test at both T0 and T20, with higher maximum oxygen uptake and maximum workload as well as longer time to exhaustion. In all three groups, peripheral blood lactate (3 min after exhaustion) was slightly higher at T20 compared to T0 without any significant intra-group differences (Table 2).

### 3.5. Body Composition

Body composition parameters at T0 and T20 are shown in Table 3 and Appendix A. Due to a high correlation between FM estimates derived from DXA and BIA (Appendix B: Figure A1), in the following it is only referred to the DXA-derived FM estimates. The KD showed significantly lower BMI as well as overall and visceral FM than the other groups already at T0. These differences were qualitatively maintained until T20. Notably, body weight (BW) and overall and visceral FM were further reduced in the KD group during the intervention (although these changes were not statistically significant), but SMM remained fairly stable. The SD group also lost BW and reduced their FM, although these changes were not nominally significant either. The most prominent changes in body composition were achieved by the LCD group, which significantly reduced BW, BMI, overall and visceral FM, as well as SMM and body cell mass (BCM) (Table 3).

BMD and PA were similar in all diet groups (KD/LCD/SD) at T0 and remained stable until T20 (Table 3 and Appendix A).

All three diet groups improved their SMM/FM ratio based on a more pronounced reduction of FM (median 6.4% in KD; 9.8% in LCD; 6.0% in SD group) compared to SMM (0% in KD; 1.8% in LCD and 0.6% in SD group) during the intervention.

### 3.6. Quality of Life (QoL)

QoL, assessed using the EORTC-QLQC30 questionnaire, improved in all three diet groups. At T0, the KD group had higher physical functioning and lower fatigue scores than the other two groups, resulting in a significant inter-group difference (Table 4).

The improvement in fatigue was not significant in the KD group upon intervention; however, overall QoL improved significantly (*p* = 0.004) and remained the highest among the diet groups. The KD group also achieved improvements in emotional functioning (*p* = 0.006) and insomnia (*p* = 0.01), resulting in significant differences among the three groups at T20. Participants in the LCD group significantly improved in eight of the 14 parameters of QoL, and SD participants improved in two parameters (physical and social functioning). Symptoms of dyspepsia (nausea/vomiting, appetite loss, constipation and diarrhea) were generally slightly improved in all three diet groups.

### 3.7. Blood Parameters

Results of blood chemistry are shown in Table 5. At T0, there was a significant difference between the groups with respect to the triglyceride (TG)/high-density lipoprotein (HDL) and low-density lipoprotein (LDL)/HDL ratio, which was lowest in the KD group. The LCD group achieved a significant reduction in their TG and LDL cholesterol levels and a non-significant increase in HDL cholesterol. However, the final median TG/HDL ratio of 1.3 did not, in either the LCD or SD group, reach the recommended ratio of <1.25. That was only achieved in the KD group (median 0.9). Due to the distinct changes in LDL and HDL, the median LDL/HDL ratio was nearly the same in all three groups at T20 (1.8–2.0).

Throughout the intervention, no significant change occurred in blood glucose and insulin levels in the KD and SD groups. Only the LCD group significantly reduced their glucose and insulin levels, and subsequently the homeostatic model assessment of insulin resistance index (HOMA-IR) index. Insulin-like growth factor -1 (IGF-1) levels did not change substantially in either group.

Concerning markers of kidney function, all three intervention groups started from comparable ranges. While the KD and SD group displayed no significant effect of diet type on creatinine, glomerular filtration rate (GFR), and uric acid in, the LCD group experienced a significant decrease in creatinine, an almost significant reduction in uric acid (*p* = 0.008), and a significant increase in GFR. Bound urea nitrogen increased significantly in the LCD group, and almost significantly (*p* = 0.008) in the KD group.

The LCD group was also the only group to experience significant changes in liver parameters; alkaline phosphatase (AP) and aspartate transaminase (AST) decreased by a median of 4 and 1 U/l, respectively.

At T0 and T20, C-reactive protein (CRP) was lowest in the KD group without significant differences within or between groups.

## 4. Discussion

The aim of this study was to test three different diets for breast cancer patients (KD, LCD, SD) with respect to their safety, feasibility, and measurable physiological and psychological effects. All diets were well tolerated and safe and supported the quality of life and physical performance of BC patients in the rehabilitation process, although most significant improvements were observed within the LCD group.

The excellent compliance of the subjects during the outpatient phase is proven by the analysis of food diaries, the match between the measured respiratory quotient with that expected for a given diet, and changes in serum parameters (especially the TG/HDL ratio), which are in line with previously published data obtained for LCDs and KDs [32] compared to SDs.

Muscle loss [33] and increased fat mass (in particular visceral fat [34]) are negatively associated with the outcome of breast cancer patients. Several adipose tissue-mediated mechanisms including immune dysregulation, chronic systemic inflammation, and elevated growth factors provide evidence that these correlations are causal [35,36]. Here, all three diet groups were able to reduce body weight, which was one of their personal goals since their physicians told them that normalizing body weight might improve outcome, based on international guidelines [37]. The avoidance of refined carbohydrates was a commonality among all three diet types and could have played a key role in inducing weight loss, on the one hand by reducing secretion of the “fattening” hormone insulin [38], which indeed was lower at T20 than at T0 in all diet groups, and on the other hand by promoting a healthy microbiome [39]. Weight loss consisted mainly of fat mass. Accordingly, the muscle/fat mass ratio improved in all groups, whereby the KD group had a higher ratio at both T0 and T20. This finding is in accordance with previously published beneficial effects of a KD in reducing central obesity and SMM preservation in female cancer patients [40,41]. LCD patients, who started from the worst ratio, achieved the most significant improvement. This indicates that a protein and fat enriched LCD could be a good choice for cancer patients wanting to improve their body composition without the need to be as strict as patients on the KD, which is in line with a recent pilot study investigating the effects of a low carbohydrate Paleolithic diet in breast cancer patients during radiotherapy [42]. Very recently, the weight loss benefits of a LCD were also shown in overweight men with prostate cancer [43]. A reason for the superiority of fat-rich diets could be that they are better adapted to the increased fat oxidation rates found in cancer patients [7,8]; the latter correlates with a decreased ability to use glucose in the periphery due to insulin resistance [5]. In this study, the KD was the only regimen that covered the recommendations for energy and protein intake during chemotherapy and radiation [44] and thus could be considered as a possible supportive diet during cancer treatment in such cases.

The RQ is a strong indicator of fuel utilization. Due to the prevalent pre-existing adaption to a LCD within the KD group, the RQ was the lowest already at T0. However, the RQ declined further until T20, possibly reflecting the influence of a well-designed KD calculated by the trained dietitian compared to the self-prescribed LCDs consumed before the intervention. This decline in the RQ reflected a further shift in fuel utilization towards fat oxidation, in line with the findings of previous studies in obese [45] or lean [46] subjects after at least three weeks on a KD. Maximum RQ values remained stable in both the LCD and SD groups, so that a significant between-group difference was observed at T20.

It is interesting to note, however, that patients reached similar lactate concentrations at T20 compared to T0, despite the different nature of the diets. This indicates that glycogen stores were not fully depleted during the exercise protocol even in the KD group and that adaption to a KD did not impair glycolysis and performance during high-intensity exercise. VO_2_/kg (max) values are frequently used in the literature to compare exercise capacity after an intervention [47]. Increases in VO_2_/kg (max) could correlate partially to the decreases in body weight of the patients and thus alone did not necessarily confirm an increase in physical performance. As workload and time to exhaustion did not change significantly in the KD group, the increase in VO_2_/kg (max) indeed appears to reflect the effects of weight loss rather than improvements in exercise capacity. However, it is noteworthy that the KD participants ended up with V02max/kg comparable to healthy women of similar age [48]. This indicates that high amounts of carbohydrates in the diet were not necessary for the fitness demands that were placed on the study participants.

The positive impact of the three-week inpatient rehabilitation program used in this study on QoL of breast cancer patients has already been published [49]. In this previous study, no special dietary intervention was used, but the other parameters of the comprehensive multi-modal program were the same as in the KOLIBRI study. All three diet groups reached or outreached the mean QoL values achieved in the previously published papers that did not include a dietary intervention at the end of the fist inpatient phase (Table 4; Appendix A). It can therefore be speculated that any type of nutritional counselling should be included in rehabilitation programs to further improve results. Functional scales remained relatively stable during the outpatient phase on all three diets. However, in parallel to the increase in physical performance, the KD group had the best physical functioning value, reaching the mean value of healthy persons published as the “reference scale” [50]. This positive effect of KD on physical functioning was also seen in women with endometrial and ovarian cancer [51]. Further, a clear discrepancy was seen in the development of some of the symptoms amongst the groups during the outpatient phase (T3-T20). While fatigue and insomnia scores remained stable in the LCD and SD groups, they further improved in the KD group to T20, almost reaching the reference values of healthy age-matched adults. This remarkable reduction in fatigue was also seen in other cancer patients eating a KD [51].

Recently, the KETOCOMP study also showed various improvements in QoL parameters in women with early-stage breast cancer consuming a KD during radiotherapy [52]. However, a recent Iranian study showed that KD did not significantly improve QoL in more advanced breast cancer patients over a 12-week interval during chemotherapy [53]. Contrary to the patients in the KOLIBRI trial, which significantly reduced their fatigue score, the Iranian patients clearly increased in fatigue, presumably due to the chemotherapy, which might have counteracted the beneficial effects of a KD on QoL observed here. It is noteworthy that the KDs have shown promise also outside the oncological context for improving mood, physical functioning, and fatigue, effects that are thought to originate partly from anti-inflammatory and anti-oxidative properties of ketone bodies, especially within the brain [54,55].

Although evidence exists that fat-rich diets do improve rather than worsen cardiovascular risk factors and especially increase HDL and decrease triglycerides [32,56,57], some oncologists fear that such diets have a detrimental effect on blood lipid profiles and liver and kidney function. Here, a remarkable positive effect of the LCD on nearly all parameters could be observed—a significant improvement in lipid profile and optimization of the liver profile and kidney function parameters. Together with the positive effects on body composition and physical performance, this would speak in favor of a LCD as a recommendable diet for patients with metabolic syndrome, such as in non-cancer patients in general [58] or BC patients in particular who are frequently overweight and could benefit from reducing their body weight and improving their metabolic health [59] and inflammatory situation [60]. Apart from a minimal increase in LDL, KD patients had the highest HDL cholesterol and lowest triglycerides of all groups with no out-of-reference values in any other parameters, indicating the safety of a KD. In addition, KD patients had low insulin concentrations, similar to those observed in healthy persons [61]. Taking into account that insulin is discussed as being a growth factor in neoplasia [35,62], the KD appears to be the most effective strategy to attenuate insulin-mediated growth stimuli to tumor cells.

It is noteworthy that the effects on biochemical parameters were more prominent in the long-term intervention (T20) compared to the short-term intervention at T3 (Appendix A), again pointing out the advantage of a 20-week dietary intervention compared to short-term interventions that may not allow for a full judgement of profound metabolic changes.

Based on the known metabolic situation in cancer patients, both the KD and the LCD can be expected to better account for the metabolic alterations, compared to the SD regimen recommended thus far. In addition, the KD was reported to be safe for cancer patients in several clinical settings [40,41,51,63] and to improve overall survival in a group of breast cancer patients compared to a SD [64].

This study has several limitations. The most obvious is the lack of randomization. This introduced bias due to the pre-existing experience of the majority of KD patients with LCDs or even KDs for a long period of time prior to T0 and an unbalanced distribution of several baseline variables between groups. For this reason, no intergroup comparison was conducted, which prohibits claims about the superiority or inferiority of any specific diet above the others. The KD group was highly motivated to participate in the study in order to maintain their diet throughout the rehabilitation procedure. To ensure maximal compliance, patients were allowed to choose their diet since the realization of the diets throughout the outpatient phase required a high level of personal motivation and active participation (food preparation, calculation). Further, due to containing the highest rate of advanced cancer patients, the KD group had the highest impetus to achieve good performance in the study.

Another point for critique could be the lack of control regarding the amount of exercise in the outpatient phase. We cannot rule out the possibility that some participants increased their training volume and/or intensity during this phase, although subjects affirmed this not to be the case. Nevertheless, the results indicate that for all subjects, physical performance improved during the intervention and was not compromised by the low (LCD) or very low (KD) carbohydrate content of their diet. This is in accordance with published data from healthy subjects and athletes, who also performed well under LCD and KD regimes [65,66,67,68,69]. In principle, lowering carbohydrate intake could lead to intramuscular glycogen depletion and subsequent impairment of maintaining higher exercise workloads. For example, Klement et al. have found that a KD in breast cancer patients induced a rapid loss of glycogen-bound water occurring within the first week after initiation [41]. While some studies showed a negative impact when switching to LCDs or KDs on physical performance in the short term (21–30 days; [70]) and up to 10 weeks [71], the duration of 20 weeks in this study apparently allowed the patients to adapt to the profound change in energy metabolism. Accordingly, at T20, none of the KD and LCD patients reported a “lack of energy”, and their physical performance was even superior to patients in the SD group. However, it must be noted that not all patients had participated in the final physical performance examination since in five cases (four KD; one SD), health insurances companies refused to fund participation at week T20.

Strengths of the study include the well-controlled dietary intervention, the uniform multimodal treatment of patients within the closed setting of the rehabilitation center, and a tight supervision in the outpatient phase by the study team (physician, dietitian, study nurse). Furthermore, the 20-week intervention ensured a profound adjustment of fuel utilization according to diet type and was longer than in typical comparable trials comprising at least 12 weeks of observational time only [53]. This resulted in measurements at T20 that were free of problems with metabolic adaption. The diet effects were reflected in both objective parameters (serum analysis, spiroergometry, DXA, and BIA) and in patient-reported outcomes determined by validated questionnaires.

Thus, this work provides further evidence supporting the safety and benefits of this dietary approach for BC patients.

## 5. Conclusions

Normal BMI, a positive body composition, and physical performance are, alongside the effects of local and systemic therapies, important predictors of breast cancer outcome. Oncologists therefore often encourage their patients to normalize their body weight by eating a healthy diet and increasing physical activity. Fat-rich low-carbohydrate diets (LCDs) and ketogenic diets (KDs) are able to induce profound positive changes in body weight and composition. The question of whether LCDs and KDs are safe and compatible with physical performance and quality of life of cancer patients can be answered with a “yes” for BC patients during their rehabilitation process, according to the data of the KOLIBRI trial. Here, it was demonstrated that both a well-defined KD as well as a less strict LCD exhibited positive effects on body composition, physical performance, and quality of life of BC patients during the rehabilitation phase. In contrast, a standard diet (SD) following the current guidelines of nutritional societies was inferior for physical performance and body composition improvements and also failed to meet the recommended energy and protein intake for cancer patients. A reason could be that SD guidelines are intended for healthy persons, who have other metabolic needs than cancer patients.

## Figures and Tables

**Figure 1 nutrients-13-01029-f001:**
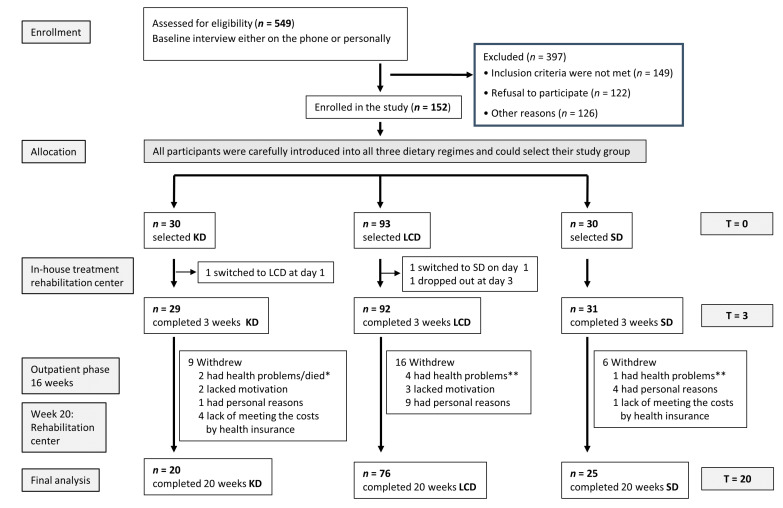
Flow chart of the study. This analysis focuses on a comparison between week 0 (T0) and week 20 (T20). * 1 patient had to resume chemotherapy directly after T3 due to progress, one died because of advanced metastatic disease. ** Patients reported health problems without further specification. KD: ketogenic diet; LCD: low carb diet; SD: standard diet.

**Figure 2 nutrients-13-01029-f002:**
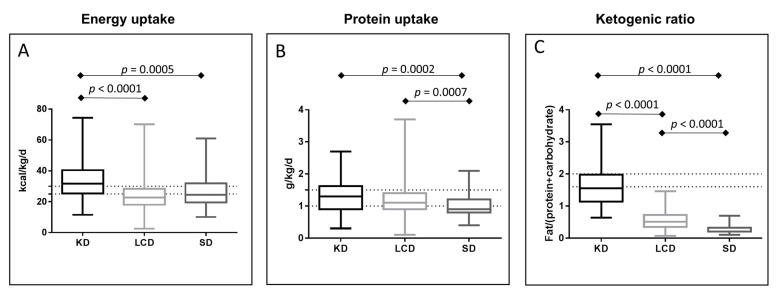
Diet analysis. Energy intake (**A**), dashed lines: optimal daily energy intake (25–30 kcal/kg/d) as recommended for ambulatory patients [31]; protein intake (**B**), dashed lines: goal for daily protein intake in cancer treatment (1–1.5 g/kg/day) according to [31]; and ketogenic ratio (**C**), defined as the amount of fat divided by the sum of protein and carbohydrates, dashed lines: goal in the study 1.6–2.1.

**Table 1 nutrients-13-01029-t001:** Baseline characteristics of the patients comprising the three intervention groups.

Parameter	Ketogenic Diet KD (*n* = 29)	Low Carb Diet LCD (*n* = 92)	Standard Diet SD (*n* = 31)	*p*-Value
Age (year)	53 (38–64)	52 (26–66)	53 (37–60)	n.s.
Karnofsky index	100 (90–100)	100 (80–100)	100 (80–100)	n.s.
-Body composition				
Body mass index (kg/m^2^)	23.4 (18.1–35.4)	27.2 (18.0–41.0)	26.6 (17.6–40.2)	0.0010 *
Fat mass (kg)	20.1 (10.3–41.5)	27.8 (6.7–55)	23.4 (9.3–40.7)	0.0006 *
Visceral fat mass (kg)	8.2 (4.0–24.0)	13.6 (2.9–28.4)	13.1 (4.2–23.4)	0.0006 *
Skeletal muscle mass (kg)	41.7 (35–49.4)	42.2 (32.5–56.7)	38.8 (29.9–54)	0.365
Phase angle (°)	5.68 (4.36–6.94)	5.55 (4.43–6.61)	5.8 (4.51–6.53)	0.25
Menopause				n.s.
Premenopause	5 (17.2%)	22 (23.9%)	10 (32.3%)	
Postmenopause	14 (48.3%)	54 (58.7%)	16 (51.6%)	
Unknown	10 (34.5%)	16 (17.4&)	5 (16.1%)	
Neoadjuvant chemotherapy				n.s.
No	20 (69.0%)	75 (81.5%)	26 (83.9%)	
Yes	9 (31.0%)	17 (18.5%)	5 (16.1%)	
Metastases				0.0031 *
No	19 65.5%)	86 (93.4%)	27 (87.1%)	
Yes	6 (20.7%)	3 (3.3%)	2 (6.45%)	
Unknown	4 (13.8%)	3 (3.3%)	2 (6.45%)	
Estrogen receptor status				n.s.
Negative	8 (27.6%)	14 (15.2%)	3 (9.7%)	
Positive	20 (69.0%)	78 (84.8%)	28 (90.3%)	
Unknown	1 (3.4%)	0	0	
Progesterone receptor status				n.s.
Negative	9 (31.0%)	15 (16.3%)	4 (12.9%)	
Positive	19 (65.5%)	77 (83.7%)	27 (87.1%)	
Unknown	1 (3.4%)	0	0	
HER2/neu status				n.s.
Negative	23 (79.3%)	77 (83.7%)	23 (74.2%)	
Positive	6 (20.7%)	15 (16.3%)	8 (15.8%)	
Anti-Hormone Therapy				n.s.
Tamoxifen	10 (34.5%)	54 (58.7%)	19 (61.3%)	
Aromatase Inhibitor	6 (20.7%)	23 (25.0%)	6 (19.35%)	
None	13 (44.8%)	15 (16.3%)	6 (19.35%)	
Herceptin				n.s.
Yes	5 (17.2%)	13 (14.1%)	6 (19.4%)	
No	24 (82.8%)	79 (85.9%)	25 (80.6%)	

Continuous variables are given as median and range and categorical variables as absolute and relative frequencies. The null hypothesis of no differences between the three groups was tested using the Kruskal–Wallis test and Fisher’s exact test for continuous and categorical variables, respectively. * *p* < 0.005 (statistically significant difference). n.s.: not statistically significant.

**Table 2 nutrients-13-01029-t002:** Performance parameters at start and end of intervention.

	KD (*n* = 19)			LCD (*n* = 70)			SD (*n* = 23)		
Parameter and Unit	T0	T20	*p*-Value	T0	T20	*p*-Value	T0	T20	*p*-Value
RQ	0.82 (0.63–0.91)	0.75 (0.65–0.80)	n.s.	0.83 (0.70–1.07)	0.85 (0.68–1.06)	n.s.	0.90 (0.74–1.01)	0.90 (0.77–1.07)	n.s.
VO_2_/kg(max) (mL/(kg × min))	22.4 (13.7–36.0)	27.9 (16.7–38.3)	n.s.	20.0 (8.4–34.6)	22.2 (13.0–36.8)	0.0003 *	19.8 (11.6–28.9)	22.5 (10.2–38.2)	0.0013 *
Threshold power (Watt)	117 (75–178)	121 (87–185)	n.s.	111 (50–175)	113 (65–161)	n.s.	93 (50–148)	90 (49–160)	n.s.
Maximum power (Watt)	140 (75–218)	139 (105–209)	n.s.	126 (68–178)	130 (70–186)	n.s.	121 (65–166)	128 (72–170)	n.s.
TTE (min)	8.33 (4.17–13.7)	8.17 (5.83–13.0)	n.s.	7.59 (3.67–11.0)	7.67 (3.67–11.5)	n.s.	7.17 (3.0–10.2)	7.67 (3.83–10.3)	n.s.
Lactate (mmol/L)	3.77 (1.66–6.04)	4.47 (2.20–8.04)	n.s.	3.94 (0.58–6.30)	4.03 (0.91–8.90)	n.s.	3.55 (1.60–5.09)	4.28 (1.88–6.34)	n.s.

KD: ketogenic diet, LCD: low carb diet, SD: standard diet; RQ: Respiratory quotient; TTE: Time to exhaustion; * *p* < 0.005 (statistically significant). n.s.: not significant.

**Table 3 nutrients-13-01029-t003:** Body composition.

		KD			LCD			SD		T0 Intergroup
	T0	T20	*p*-Value	T0	T20	*p*-Value	T0	T20	*p*-Value	*p*-Value
Weight (kg)	65.2 (49.8–94)	62.9 (49.1–86.1)	0.009	74.1 (48.9–116.2)	69.2 (46.4–115.1)	<0.0001 *	68.6 (42.8–97.9)	65.8 (43–93)	0.012	n.s.
BMI (kg/m^2^)	23.4 (18.1–35.4)	22.1 (17.8–32.4)	0.009	27.2 (18.0–41)	25.1 (17.5–42)	<0.0001 *	26.6 (17.6–40.2)	25 (17.7–36.8)	0.0098	0.001 *
FM (kg)	20.1 (10.3–41.5)	17.5 (11.3–36.8)	n.s.	27.8 (6.7–55)	24.2 (8.4–41.6)	<0.0001 *	26.9 (10.4–40)	23.4 (9.3–40.7)	0.017	0.0006 *
Visceral FM (kg)	8.2 (4.0–24.0)	6.9 (4.1–20.2)	n.s.	13.6 (2.9–28.4)	11.3 (3.6–31.0)	<0.0001 *	13.1 (4.2–23.4)	11.7 (4.1–22.8)	0.031	0.0006 *
SMM (kg)	41.7 (35–49.4)	40.8 (34.2–47.8)	n.s.	42.2 (32.5–56.7)	41.3 (33.3–58.8)	0.0011 *	38.8 (29.9–54)	38.5 (31.2–54.6)	0.200	n.s.
SMM/FM	2.0 (1.2–4.3)	2.4 (1.3–3.7)	n.s.	1.5 (0.9–6.4)	1.6 (1.0–5.2)	<0.0001 *	1.6 (1.1–2.9)	1.7 (1.1–3.4)	0.037	<0.0001 *
BIA (BCM) (kg)	23.1 (18.1–26.2)	21.7 (18.7–26.8)	n.s.	22.7 (16.0–31.2)	22.4 (16.0–31.2)	<0.0001 *	22.7 (15.6–30.3)	21.8 (15.6–30.5)	0.217	n.s.
BIA PA (°)	5.68 (4.36–6.94)	5.62 (4.61–6.91)	n.s.	5.55 (4.43–6.61)	5.59 (4.36–6.73)	n.s.	5.8 (4.51–6.56)	5.7 (4.61–6.53)	0.726	n.s.
BMD (g/cm^2^)	1.10 (0.88–1.43)	1.10 (0.87–1.28)	n.s.	1.07 (0.88–1.49)	1.06 (0.87–1.52)	n.s.	1.04 (0.84–1.31)	1.05 (0.92–1.26)	0.361	n.s.

KD: ketogenic diet, LCD: low carb diet, SD: standard diet, BMI: body mass index, FM: fat mass, SMM: skeletal muscle mass, BCM: body cell mass, PA: phase angle alpha, BMD: bone mineral density, * *p* < 0.005 (statistically significant); n.s.: not statistically significant.

**Table 4 nutrients-13-01029-t004:** Quality of life (EORTC-QLQ C30, version 3).

		KD (*n* = 20)			LCD (*n* = 75)			SD (*n* = 24)		T0 Inter-Group
	T0	T20	*p*-Value	T0	T20	*p*-Value	T0	T20	*p*-Value	*p*-Value
Glob. health/QoL	66.7 (25–92)	75 (16.7–100)	0.004 *	50 (0–83,3)	66.7 (16.7–100)	<0.0001 *	54.2 (16.7–83.3)	66.7 (25–91.7)	0.008	n.s.
Physical funct.	80 (40–100)	93.3 (20–100)	n.s.	73.3 (20–100)	80 (46.7–100)	<0.0001 *	73.3 (40–100)	83.4 (46.7–100)	0.002 *	0.0005 *
Emotional funct.	58.3 (0–100)	79.2 (0–100)	0.006	50 (0–100)	66.7 (0–100)	<0.0001 *	50 (0–100)	58.3 (0–100)	n.s.	n.s.
Cognitive funct.	66.7 (0–100)	75 (16.7–100)	n.s.	66.7 (0–100)	66.7 (0–100)	n.s.	58.4 (0–100)	66.7 (16.7–100)	n.s.	n.s.
Social funct.	66.7 (0–100)	66.7 (0–100)	n.s.	66.7 (0–100)	66.7 (0–100)	0.002 *	66.7 (0–100)	66.7 (33.3–100)	0.002 *	n.s.
Role funct.	66.7 (0–100)	75 (0–100)	n.s.	50 (0–100)	66.7 (0–100)	<0.0001 *	66.7 (0–100)	66.7 (16.7–100)	n.s.	n.s.
Fatigue	33.3 (0–100)	11.1 (0–88.9)	n.s.	66.7 (0–100)	33.3 (0–100)	<0.0001 *	55.6 (0–100)	50 (0–88.9)	n.s.	<0.0001 *
Pain	33.3 (0–100)	25 (0–100)	n.s.	66.7 (0–100)	33.3 (0–100)	0.0001 *	50 (0–100)	33.3 (0–83.3)	n.s.	n.s.
Dyspnea	33.3 (0–100)	33.3 (0–100)	n.s.	66.7 (0–100)	33.3 (0–100)	<0.0001 *	33.3 (0–100)	33.3 (0–66.7)	n.s.	n.s.
Insomnia	66.7 (0–100)	33.3 (0–100)	0.010	66.7 (0–100)	66.7 (0–100)	n.s.	66.7 (0–100)	33.3 (0–100)	n.s.	0.005
Nausea/ Vomiting	0 (0–100)	0 (0–50)	n.s.	0 (0–50)	0 (0–50)	n.s.	0 (0–83.3)	0 (0–66.7)	n.s.	n.s.
Appetite loss	0 (0–100)	0 (0–66.7)	n.s.	0 (0–66.7)	0 (0–66.7)	n.s.	0 (0–100)	0 (0–66.7)	n.s.	n.s.
Consti- pation	0 (0–100)	0 (0–66.7)	n.s.	0 (0–100)	0 (0–100)	n.s.	0 (0–66.7)	0 (0–66.7)	n.s.	n.s.
Diarrhea	0 (0–100)	0 (0–66.7)	n.s.	0 (0–100)	0 (0–100)	n.s.	0 (0–66.7)	0 (0–66.7)	n.s.	n.s.

KD: ketogenic diet, LCD: low carb diet, SD: standard diet, QoL: Quality of life, funct: functioning, * *p* < 0.005. n.s.: not statistically significant.

**Table 5 nutrients-13-01029-t005:** Blood parameters at start and end of intervention.

	KD			LCD			SD			Intra-Group Diff.
Parameter and Dimension	T0	T20	*p*-Value	T0	T20	*p*-Value	T0	T20	*p*-Value	T0 *p*-Value
TG (mg/dL)	70 (45–177)	78 (42–184)	n.s.	104 (42–489)	84 (43–330)	<0.0001 *	91 (48–265)	85 (50–261)	n.s.	<0.0001 *
Chol (mg/dL)	243 (167–352)	242 (142–357)	n.s.	224 (157–373)	219 (127–338)	<0.0001 *	213 (152–328)	207 (149–256)	n.s.	n.s.
HDL (mg/dL)	76 (39–129)	74 (39–121)	n.s.	65 (37–103)	67 (39–104)	0.008	66 (33–94)	64 (44–94)	n.s.	0.0003 *
LDL (mg/dL)	141 (73–226)	157 (67–205)	n.s.	145 (87–266)	140 (76–234)	<0.0001 *	137 (93–234)	122 (88–174)	n.s.	n.s.
LDL/HDL	1.7 (0.6–4.6)	1.8 (0.7–4.6)	n.s.	2.3 (1.0–5.0)	2.0 (0.9–3.7)	<0.0001 *	2.1 (1.1–3.7)	1.9 (1.0–3.4)	n.s.	0.002 *
TG/HDL	0.9 (0.4–4.5)	0.9 (0.4–3.5)	n.s.	1.7 (0.4–9.8)	1.3 (0.4–8.1)	<0.0001 *	1.4 (0.6–6.1)	1.3 (0.6–5.2)	n.s.	<0.0001 *
Glucose (mg/dL)	85 (67–114)	86 (64–104)	n.s.	89 (72–161)	87 (74–132)	0.004 *	84 (68–100)	86 (74–109)	n.s.	n.s.
Crea (mg/dL)	0.81 (0.59–1.1)	0.80 (0.6–1.0)	n.s.	0.76 (0.51–1.1)	0.72 (0.5–1.0)	0.0002 *	0.77 (0.56–1.2)	0.80 (0.51–1.1)	n.s.	n.s.
GFR (mL/min)	76.6 (55.4–110)	85.1 (64–106)	n.s.	91.2 (53.6–125)	95.6 (60–113)	0.0002 *	90.9 (49.4–115)	84.8 (56.7–122)	n.s.	n.s.
Uric acid (mg/dL)	4.8 (2.7–7.6)	4.8 (3.6–6.6)	n.s.	5.2 (2.8–8.8)	4.9 (2.8–8.0)	0.008	5.1 (3.4–8.7)	4.9 (3.2–9.2)	n.s.	n.s.
BUN (mg/dL)	30 (24–47)	32 (25–46)	n.s.	28 (16–47)	31 (19–61)	<0.0001 *	29 (13–41)	31 (18–40)	n.s.	n.s.
AP (U/L)	62 (26–163)	63 (29–111)	n.s.	66 (26–120)	62 (32–156)	0.003 *	55 (24–125)	64 (24–123)	n.s.	n.s.
AST (U/L)	20 (10–126)	21 (10–85)	n.s.	23 (9–160)	22 (12–96)	0.0049 *	22 (10–64)	22 (13–60)	n.s.	n.s.
CRP (mg/L)	0.53 (0.06–6.7)	0.65 (0.2–10.3)	n.s.	1.7 (0.02–10.3)	1.0 (0.1–10.4)	0.007	2.0 (0.03–12.2)	1.2 (0.2–7.7)	n.s.	n.s.
TSH (mU/L)	1.3 (0.4–3.7)	1.4 (0.4–5.2)	n.s.	1.4 (0.1–8.9)	1.5 (0.06–4.5)	n.s.	0.8 (0.05–3.6)	0.9 (0.3–3.7)	n.s.	0.004 *
Insulin (µU/L)	10.2 (5–33.3)	9.6 (6.3–31.6)	n.s.	14.6 (4.0–45.1)	13.0 (5–45.)	0.001 *	13.9 (7.3–60.4)	12.8 (5.9–41.6)	n.s.	n.s.
IGF-1	9.1 (4.1–50)	9.0 (2.7–48)	n.s.	7.0 (1.5–19.4)	6.5 (1.2–18)	n.s.	5.7 (1.7–22.4)	6.2 (2.3–21.9)	n.s.	n.s.
HOMA-IR	2.1 (0.8–7.6)	2.0 (1.3–6.8)	n.s.	3.3 (0.8–10.5)	2.7 (1.0–10.2)	<0.0001 *	2.7 (1.5–14.2)	2.8 (1.1–9.1)	n.s.	n.s.

KD: ketogenic diet, LCD: low carb diet, SD: standard diet; TG: triglycerides; Chol: total cholesterol; HDL: high-density lipoprotein; IGF-1: Insulin-like growth factor-1LDL: low-density lipoprotein; TG: triglyceride; Crea: creatinine; GFR: glomerular filtration rate; BUN: bound urea nitrogen; AP: alkaline phosphatase; AST: aspartate transaminase; CRP: C-reactive protein; TSH: thyoid-stimulating hormone; HOMA-IR: homeostatic model assessment of insulin resistance index: 1.9–2.9 = early insulin resistance; >2.9: significant insulin resistance. Blood parameters for the three intervention groups measured at T0 and T20. Parameter values are given as median (range). * *p* < 0.005 (statistically significant); n.s.: not statistically significant.

## Data Availability

All patient data are available anonymized in a local database. Requests to access the datasets should be directed to frak057@mail.uni-wuerzburg.de.

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
