# Peer review of "Low Carb and Ketogenic Diets Increase Quality of Life, Physical Performance, Body Composition, and Metabolic Health of Women with Breast Cancer"

_nutrients, 2021, doi:10.3390/nu13031029_

Round 1

Reviewer 1 Report

The manuscript explored to the effects of a low-carbohydrate diet and a KD diet during the rehabilitation period for breast cancer patients. The meaningful contents included for improving the prognosis QOL of breast cancer patients though, the analysis method requires major revisions.

Major Comments

/ Patient baseline characteristics showed some significant differences, especially in the KD diet group, and many in the KD diet group had already experienced LC dietary restrictions. Three groups with different backgrounds (BMI, metastases , anti-hormone therapy, and maybe FM and visceral FM) of the subjects should not be compared the effect of dietary intervention. Nonetheless, the authors determine that the KD diet is beneficial for BC patient compared to SD diet.

This study should be divided into two studies; the study 1 comparing SD diet and LC diet with no background difference, and the study 2 examining only the effect of KD diet. The discussion also needs to be rewritten in two separate studies.

/ Many studies has been reported about the effect of ketogenic and low-carbohydrate diets to breast cancer. Clarify what is the point of originality in your study compared to previous studies. Previous studies should be added to the discussion and compared

Specific Comments

  • Why use the p-value p <0.005 to define the statistical significance, instead of the general value p <0.05.

  • Table 1 should show body composition values such as Fat mass as body composition along with BMI. Baseline Fat mass might be significantly lower in the KD group.

  • In Table 3 and Table 4, the item has a line break in the middle. Should be recreated the table for the reader to easy understand, such as using the page horizontally or widening the width of the table.
  • Same data were shown in Table 3 and Figure 3, so you should delete Figure 3.
  • the blood data statistics in Table 5 were correct? In particular, glucose in the LC diet group decreased by only 1 mg / dL, but p = 0.004, was it true ?

  • In Figure 2, dietary protein intake is higher in the LC group than in the KD group. However, the KD value (Figrue 2C) is considerably higher in the KD group. I doubt this is correct. Enter a description in the legend as to whether the bar results are mean and range or SD. The dietary intake data might be understood easier by simply comparing the PFC energy ratio.

Author Response

Thank you very much for the valuable comments. Following your  suggestions, we have thoroughly revised our manuscript. Please find the point by point answers below:

Reviewer 1

Major Comments

/ Patient baseline characteristics showed some significant differences, especially in the KD diet group, and many in the KD diet group had already experienced LC dietary restrictions. Three groups with different backgrounds (BMI, metastases , anti-hormone therapy, and maybe FM and visceral FM) of the subjects should not be compared the effect of dietary intervention. Nonetheless, the authors determine that the KD diet is beneficial for BC patient compared to SD diet.

  1. This study should be divided into two studies; the study 1 comparing SD diet and LC diet with no background difference, and the study 2 examining only the effect of KD diet. The discussion also needs to be rewritten in two separate studies.

Answer: Following this comment and according to different number of participants in the LOGI group compared to SD and KD, we decided to better rate the three groups in parallel now omitting the intergroup comparison and to change the discussion in the direction that all three interventions improved the patients’ outcome. However, the largest improvements were seen with the LCD and the KD group who remained on the best levels concerning fitness and several blood chemistry markers

  1. Many studies have been reported about the effect of ketogenic and low-carbohydrate diets to breast cancer. Clarify what is the point of originality in your study compared to previous studies. Previous studies should be added to the discussion and compared.

Answer: We have added some very recent studies on the ketogenic diet in breast cancer patients. However, the originality in our study is the comparison of three diet groups in a very controlled rehabilitation setting and the readout parameters like body composition analyzed in two well established settings together with physical fitness by spiroergometry, quality of life and blood parameters all analyzed in parallel. Further, our observation time of 20 weeks extends that of the common 12 weeks in other trials.

Specific Comments

  1. Why use the p-value p <0.005 to define the statistical significance, instead of the general value p <0.05.

Answer: The classical p-value threshold of 0.05 is only consistent with weak evidence against the null hypothesis when converted to minimum Bayes factors (e.g. Held L, Ott M. On p -Values and Bayes Factors. Annu Rev Stat Its Appl. 2018;5:393–419). Often such results are therefore not reproducible. To avoid this, and because of the many tests performed, we use a lower threshold of 0.005 which corresponds to strong evidence against the null hypothesis within a likelihood- or Bayes factor-based conception of evidence. Bayes factors (or likelihood ratios in case of simple hypotheses) measure the strength of evidence between two competing hypotheses (Klement RJ & Bandyopadhyay PS. Emergence and evidence: A close look at Bunge’s philosophy of medicine. Philosophies. 2019; 4(3):50).

We have added a paragraph on this aspect into the discussion section

  1. Table 1 should show body composition values such as Fat mass as body composition along with BMI. Baseline Fat mass might be significantly lower in the KD group.

Answer: We have added baseline fat mass and phase angel in Table 1 now.

  1. In Table 3 and Table 4, the item has a line break in the middle. Should be recreated the table for the reader to easy understand, such as using the page horizontally or widening the width of the table.

Answer: We reformatted the Tables 3 and 4, and since we did no longer perform intergroup-comparison at T20, one column was deleted thus giving more space for the other columns.

  1. Same data were shown in Table 3 and Figure 3, so you should delete Figure 3.

Answer: We transferred figure 3 to the supplemental material

  1. the blood data statistics in Table 5 were correct? In particular, glucose in the LC diet group decreased by only 1 mg / dL, but p = 0.004, was it true?

Answer: We checked the p-values again, and they are correct. We had performed a paired Wilcoxon test within the groups. The change indeed appears small, but it was a consistent decrease within each patient. If we would have performed a not paired test instead, the p-value would have been 0.252. However, the initial median glucose value was 89 mg/dl, not 88, so the median change was 2 mg/dl. To avoid confusion, we now omitted all non-significant p-values from the tables 2-5.

  1. In Figure 2, dietary protein intake is higher in the LC group than in the KD group. However, the KD value (Figrue 2C) is considerably higher in the KD group. I doubt this is correct. Enter a description in the legend as to whether the bar results are mean and range or SD. The dietary intake data might be understood easier by simply comparing the PFC energy ratio.

Answer: this is no contradiction: since the KD value is defined as the amount of fat compared to that amount of protein plus CHO, the LCD which had a higher protein intake + a higher CHO intake than the nearly CHO free and moderate protein KD must result in a  lower KD value in the LCD.

Reviewer 2 Report

Comments to the Author:

I thank to the editors for the opportunity to review this study, beside I would also like to congratulate the authors for the made effort in their study. The present manuscript by Kämmerer et al., analyzed “Low Carb and ketogenic diets increase quality of life, physical performance, body composition and metabolic health of women with breast cancer better than a standard diet”. The authors attempted to compare the three diet types (healthy standard diet, ketogenic diet and low carbohydrate diet) in breast cancer patients and assess feasibility, safety and tolerability. The manuscript is interesting and covers a topic that has not been well studied. Nevertheless, some important issues need to be addressed to the study. 

  1. The authors added: “This diet is still a matter of concern and leads to a debate among oncologists and nutritionists, who expect cardiovascular side effects and loss of quality of life due to the high amount of fat”. Authors should add examples of this argument.
  2. The authors added: “A less strict but also fat-enriched diet is the LCD (low carbohydrate diet). Its idea is to keep insulin levels low in order to prevent or reduce the metabolic syndrome, which has been linked to worsening cancer outcomes [13]. The LCD allows an intake of up to 120 grams of carbohydrates per day, it is balanced in protein (20% of energy/day) and 65 rich in fat (remaining calories) [14,15]”. The authors should add an explanation explaining the reason by which this type of diet is linked to worsening cancer outcomes.
  3. The authors should attempt to provide a short introduction about the benefits of a specific diet for women with breast cancer rather than a normal diet.
  4. Could the authors justify how the significant differences between groups in metastases and body mass index values do not generate a significant gender bias for the study? This issue is a major weakness in your study (in additional to the lack of a randomization and physical activity) since authors are comparing overweight (LCD and SD diet) and normal weight (KD diet) women. If it is possible, the authors should add references.
  5. The first paragraph of the discussion should be much better structured. First it should be the main aim of the study and then the most relevant results. It is not necessary to add more information because it confuses the reader.
  6. The authors must make an enormous effort to conduct a real discussion. The authors showed their results and those of other authors, but do not provide a justification to explain the origin of these results. For instance: “Further, a clear discrepancy was seen in the development of some of the symptoms amongst the groups during the outpatient phase (T3-T20). While fatigue and insomnia scores remained stable in the LCD and SD groups, they further improved in the KD group to T20, almost reaching the reference values of healthy age-matched adults. This remarkable reduction in fatigue was also seen in other cancer patients eating a KD [42].” An explanation of this data should be added.
  7. How can the authors demonstrate that the decrease in BMI was due to the diet rather than to excessive sports practice (a variable that was not controlled)?

Minor comments.

P1-line 40: Add reference.

P2-line 48: Add reference.

P2-line 52-61: Continue in the same paragraph.

P2-line 64: intakte = intake.

P6- line 210: VO2/kg (max). This mistake is repeated throughout the manuscript.

The authors should make an effort to improve the design of table 1 and 2 and increase the size of figure 2. In table, 3, 4 and 5 authors should correct the brackets, because they are kept at the same height, becoming the graphs quite unclear. 

P14-L301-307: this paragraph adds nothing to the discussion section.

P14-L308-340: Before conclusion section.

P16-line 407: Add reference.

P16-line 409: Add reference.

Author Response

Thank you very much for the valuable comments. Following your  suggestions, we have thoroughly revised our manuscript. Please find the point by point answers below:

  1. The authors added: “This diet is still a matter of concern and leads to a debate among oncologists and nutritionists, who expect cardiovascular side effects and loss of quality of life due to the high amount of fat”. Authors should add examples of this argument.

Answer: We have added references underlining this argument

  1. The authors added: “A less strict but also fat-enriched diet is the LCD (low carbohydrate diet). Its idea is to keep insulin levels low in order to prevent or reduce the metabolic syndrome, which has been linked to worsening cancer outcomes [13]. The LCD allows an intake of up to 120 grams of carbohydrates per day, it is balanced in protein (20% of energy/day) and 65 rich in fat (remaining calories) [14,15]”. The authors should add an explanation explaining the reason by which this type of diet is linked to worsening cancer outcomes.

Answer: The worsening outcome is due to the metabolic syndrome, not to the diet, however, to express this more clear, we have  replaced “which” has been linked… with “a pathological metabolic state that has been….”

  1. The authors should attempt to provide a short introduction about the benefits of a specific diet for women with breast cancer rather than a normal diet.

Answer: Since we evaluated three diet types without judging on one especial as “starting point”, we would not extend the introduction in this respect – however we have added the trial hypothesis which expresses this aspect.

  1. Could the authors justify how the significant differences between groups in metastases and body mass index values do not generate a significant gender bias for the study? This issue is a major weakness in your study (in additional to the lack of a randomization and physical activity) since authors are comparing overweight (LCD and SD diet) and normal weight (KD diet) women. If it is possible, the authors should add references.

Answer: According to reviewer 1 we now have taken the bias of different starting conditions of our patients into account and related the three groups separately without performing the intergroup correlation at T20

  1. The first paragraph of the discussion should be much better structured. First it should be the main aim of the study and then the most relevant results. It is not necessary to add more information because it confuses the reader.

Answer: Done. We formulated the main aim and the main results.

  1. The authors must make an enormous effort to conduct a real discussion. The authors showed their results and those of other authors, but do not provide a justification to explain the origin of these results. For instance: “Further, a clear discrepancy was seen in the development of some of the symptoms amongst the groups during the outpatient phase (T3-T20). While fatigue and insomnia scores remained stable in the LCD and SD groups, they further improved in the KD group to T20, almost reaching the reference values of healthy age-matched adults. This remarkable reduction in fatigue was also seen in other cancer patients eating a KD [42].” An explanation of this data should be added.

Answer: Thank you for this suggestion. We have added mechanistic explanations for our findings in several places in the discussion.

  1. How can the authors demonstrate that the decrease in BMI was due to the diet rather than to excessive sports practice (a variable that was not controlled)?

Answer: As we have mentioned in the manuscript already the participants were frequently contacted by phone and mail and did not report an increase of their daily exercise routine

Minor comments.

P1-line 40: Add reference. done

P2-line 48: Add reference. done

P2-line 52-61: Continue in the same paragraph. done

P2-line 64: intakte = intake. Corrected

P6- line 210: VO2/kg (max). This mistake is repeated throughout the manuscript. Corrected

The authors should make an effort to improve the design of table 1 and 2 and increase the size of figure 2. In table, 3, 4 and 5 authors should correct the brackets, because they are kept at the same height, becoming the graphs quite unclear.  Tables are reformated

P14-L301-307: this paragraph adds nothing to the discussion section. This paragraph was transferred to the results section

P14-L308-340: Before conclusion section. We have reordered the discussion following the reviewers suggestion

P16-line 407: Add reference. We have added: Hyde et al.

P16-line 409: Add reference. We have added: Inglis et al.

Reviewer 3 Report

STRUCTURE

  • The manuscript is properly structured

TITLE AND ABSTRACT

  • The title or abstract should inform that the type of study

INTRODUCTION

  • Line 38: add the reference, beyond the website
  • Explain the scientific background in more depth and with more bibliographic references
  • State specific objectives
  • The hypotheses are missing
  • Do not use the first person in scientific publications. Applicable to the entire document

MATERIAL AND METHODS

Trial design

  • Description of trial design (such as parallel, factorial) including allocation ratio

Participants

  • Describe the settings and locations where the data were collected
  • The flow diagram should appear in the results section
  • Figure 1: include clarification of abbreviations KD, LCD, SD

Interventions

  • Explain the interventions for each group with sufficient details to allow replication, including how and when they were actually administered

Study size

  • How sample size was determined?

RESULTS

  • Consider to use the flow diagram here´
  • Dates defining the periods of recruitment and follow-up are missing
  • Why the trial ended or was stopped?
  • The title of the table must be above the table
  • Tables: should follow the guidelines and template provided by NUTRIENTS
  • Table 1: include clarification of abbreviations KD, LCD, SD
  • Figure 3: include clarification of abbreviations KD, LCD, SD, SMM, BMD
  • Table 3: include clarification of abbreviations KD, LCD, SD, SMM, BMD, BCM, FM…
  • Table 4: include clarification of abbreviations KD, LCD, SD, QoL
  • Line 265: add the meaning of the abbreviation TG
  • Table 5: include clarification of abbreviations KD, LCD, SD
  • Add all important harms or unintended effects

DISCUSSION

  • Add reference in line 367
  • Trial limitations, addressing sources of potential bias, imprecision, and, if relevant, multiplicity of analyses
  • Discuss the generalisability (external validity) of the trial findings
  • Add interpretation consistent with results, balancing benefits and harms, and considering other relevant evidence

REFERENCES

  • References follows the indicated style

Author Response

Thank you very much for the valuable comments. Following your suggestions, we have thoroughly revised our manuscript. Please find the point by point answers below:

TITLE AND ABSTRACT

  • The title or abstract should inform that the type of study

Answer: type of study now added into abstract

INTRODUCTION

  • Line 38: add the reference, beyond the website

Answer: we have added Matuzzi et al

  • Explain the scientific background in more depth and with more bibliographic references

Answer: we have extended the references to explain the background more stringent throughout the manuscript

  • The hypotheses are missing

Answer: We now have added our working hypothesis at the end of the introduction section

  • Do not use the first person in scientific publications. Applicable to the entire document

Answer: we now have replaced all first person aspects throughout the manuscript.

MATERIAL AND METHODS

Trial design

  • Description of trial design (such as parallel, factorial) including allocation ratio

Answer: Since all three groups were handled in parallel, we added an information on this in the 2.1. Study design paragraph. No allocation ratio is available.

Participants

  • Describe the settings and locations where the data were collected

Answer: this was already described: all participants and all data collection was performed in the rehabilitation center

  • The flow diagram should appear in the results section

Answer: is transferred to the results section now

  • Figure 1: include clarification of abbreviations KD, LCD, SD

Answer: abbreviations are given in the legend now

Interventions

  • Explain the interventions for each group with sufficient details to allow replication, including how and when they were actually administered

Answer: We have added a detailed description of the multimodal rehabilitation intervention in paragraph 2.3 now

Study size

  • How sample size was determined?

Answer: Group size was calculated as 50 per group – due to the option to choose the diet, the patients however distributed in the numbers 30-93-30

RESULTS

  • Consider to use the flow diagram here´ done
  • Dates defining the periods of recruitment and follow-up are missing

Answer: Enrollement was over 12 month and the follow-up of the last patient thus 16 weeks after this 12 month. Mentioned enrolment period in 2.2 now

  • Why the trial ended or was stopped?

Answer: Funding was given for 150 patients, those were enrolled.

  • The title of the table must be above the table

Answer: Title of table 1 is now above the table like for the other ones

  • Tables: should follow the guidelines and template provided by NUTRIENTS

Answer: we have adapted Tables to the guidelines

  • Table 1: include clarification of abbreviations KD, LCD, SD done
  • Figure 3: include clarification of abbreviations KD, LCD, SD, SMM, BMD done
  • Table 3: include clarification of abbreviations KD, LCD, SD, SMM, BMD, BCM, FM…done
  • Table 4: include clarification of abbreviations KD, LCD, SD, QoL done
  • Line 265: add the meaning of the abbreviation TG done
  • Table 5: include clarification of abbreviations KD, LCD, SD done
  • Add all important harms or unintended effects

Answer: there were neither harms nor unintended effect seen with any of the diets in any of the groups. Added information see in your discussion point 3, have added a sentence on “Few patients reported mild headache and digestive problems at the beginning in all three diet types, which were self-limiting and could not clearly assigned to one diet type“ in paragraph 3.2

DISCUSSION

  • Add reference in line 367. We have added a review (Lounghey et al)
  • Trial limitations, addressing sources of potential bias, imprecision, and, if relevant, multiplicity of analyses

Answer: since the intragroup comparison at T20 was omitted now due to reviewer one, the bias of the KD group compared to the other groups now did no longer impact on the overall result

  • Discuss the generalisability (external validity) of the trial findings

Answer: we have added several aspects in this respect throughout the discudsion

  • Add interpretation consistent with results, balancing benefits and harms, and considering other relevant evidence

Answer Since there was no harm as we clearly could demonstrate by all data, this is not a matter of the discussion However, according to the observation by the treating physicians, we have added a sentence on “Few patients reported mild headache and digestive problems at the beginning in all three diet types, which were self-limited and could not clearly assigned to one diet type“ in paragraph 3.2

Round 2

Reviewer 1 Report

All parts pointed out were corrected properly.

But some tables  fomat should be modified.

Reviewer 2 Report

For the author:

I appreciate authors’ effort. The authors have obviously spent considerable time revising the manuscript and their hard work is clearly paying off. This manuscript is drastically improved from the original submission. The message is very clear, the language is much more clean, and the issues in the first version were corrected. Besides, the authors have answered all my comments successfully.

Reviewer 3 Report

No further comments.